

# Stable isotopes of Hawaiian spiders reflect substrate properties along a chronosequence

Susan R. Kennedy[1], Todd E. Dawson[1,2] and Rosemary G. Gillespie[1]

[1] Department of Environmental Science, Policy and Management, University of California, Berkeley, Berkeley, CA, United States of America
[2] Department of Integrative Biology, University of California, Berkeley, Berkeley, CA, United States of America

## ABSTRACT

The Hawaiian Islands offer a unique opportunity to test how changes in the properties of an isolated ecosystem are propagated through the organisms that occur within that ecosystem. The age-structured arrangement of volcanic-derived substrates follows a regular progression over space and, by inference, time. We test how well documented successional changes in soil chemistry and associated vegetation are reflected in organisms at higher trophic levels—specifically, predatory arthropods (spiders)—across a range of functional groups. We focus on three separate spider lineages: one that builds capture webs, one that hunts actively, and one that specializes on eating other spiders. We analyze spiders from three sites across the Hawaiian chronosequence with substrate ages ranging from 200 to 20,000 years. To measure the extent to which chemical signatures of terrestrial substrates are propagated through higher trophic levels, we use standard stable isotope analyses of nitrogen and carbon, with plant leaves included as a baseline. The target taxa show the expected shift in isotope ratios of $\delta^{15}N$ with trophic level, from plants to cursorial spiders to web-builders to spider eaters. Remarkably, organisms at all trophic levels also precisely reflect the successional changes in the soil stoichiometry of the island chronosequence, demonstrating how the biogeochemistry of the entire food web is determined by ecosystem succession of the substrates on which the organisms have evolved.

# INTRODUCTION

Evolutionary processes are determined in large part by the ecosystems within which they take place. While connecting the processes of evolutionary biology and ecology remains a critical frontier in biological sciences (*Matthews et al., 2011*), there are few studies that demonstrate how mechanisms driving processes of evolutionary biology and ecosystem science are linked. The current study seeks to understand how organismal diversity may reflect successional shifts in soil chemistry by testing the extent to which organisms at different trophic levels reflect the properties of the substrates on which they occur.

The Hawaiian Archipelago presents a highly suitable system for studying the link between evolutionary processes and ecosystem properties. The current high islands of

Corresponding author
Susan R. Kennedy,
susanrkennedy@gmail.com

Hawaii are arranged sequentially from oldest to youngest, with Kauai, at 5.1 million years, in the far northwest, and Hawai'i Island, at <1 million years, in the southeast (*Carson & Clague, 1995*). This sequential order is a consequence of the archipelago being located on a volcanic hot spot, where magma upwelling from the earth's mantle has formed into large shield volcanoes. At the same time, the tectonic plate on which the islands are situated is moving toward the northwest such that each newly emerging island has appeared to the southeast of its next-oldest neighbor. The resultant, nearly-linear age gradient makes Hawaii an ideal *chronosequence:* a temporally varied system in which the ecosystems of the younger sites are currently developing in a manner assumed to reflect the developmental history of the older sites (*Walker et al., 2010*). Given fairly precise information on the age of formation and subsequent history across a chronosequence, these systems can provide unprecedented insights into ecosystem development. Thus, chronosequences have added significantly to our understanding of how nutrients change over time (*Vitousek, 2004*) and the impact of changes in soil nutrient availability on plants (*Wardle et al., 2008*), decomposers (*Williamson, Wardle & Yeates, 2005*; *Doblas-Miranda et al., 2008*), above-ground and below-ground ecosystem processes (*Wardle, Walker & Bardgett, 2004*), and entire arthropod communities (*Gruner, 2007*).

Hawaii has served as a chronosequence for detailed studies on the ways in which ecosystem properties and functions change over extended time (*Vitousek, 2004*). Nutrient flow and its impacts on primary producers (trees) have been well characterized in this system. Studies have examined the evolution of soils on substrates of different surficial age (300 y—4.1 Mya), controlled for elevation, climate, land use history, and canopy vegetation (*Metrosideros polymorpha*), with all minerals derived from volcanic ash. An important finding of this work was that soil nitrogen, foliar nutrient availability and productivity start off very low, increase rapidly with substrate age, peak on substrates of intermediate age (*ca.* 20,000 y) on the youngest island, and then decline rapidly on older islands before all but disappearing on the oldest (*Vitousek, Shearer & Kohl, 1989*; *Vitousek, Turner & Kitayama, 1995*; *Vitousek et al., 1997*). A more recent study found that tree height and canopy nitrogen also peak on intermediate-aged (20,000 y) substrates on the youngest island (*Vitousek et al., 2009*). Nitrogen isotopes follow a similar pattern, with foliar $\delta^{15}N$ very low at the youngest sites, increasing with substrate age, and highest at a 67,000 y site (*Vitousek, Shearer & Kohl, 1989*).

Geologic history and nutrient flow evidently have important effects on the lowest trophic level—plants—but little is known about how these effects might be propagated through higher trophic levels, i.e., higher-level consumers. At the same time, work on the effects of substrate age on above-ground systems, including whole communities, has shown that community traits such as population species diversity (*Gillespie & Baldwin, 2010*; *Lim & Marshall, 2017*), genetic structure (*Roderick et al., 2012*), and network modularity (*Rominger et al., 2016*) change in a predictable manner across the Hawaiian chronosequence. However, although these community-level studies have markedly enhanced our understanding of the changes in community ecology over time, there has as yet been no attempt to link chemical changes during the evolution and development of soils (and associated ecosystem properties) with the abundance, diversity, and evolutionary

histories of above-ground organisms. The current study begins to address this question by testing the effects of substrate age on the biochemistry—isotopic signatures—of secondary consumers (predators) across a Hawaiian chronosequence.

The application of stable isotope information has revolutionized studies of nutrient flow and niche ecology in a wide range of organisms (e.g., *Fry, 1988*; *Hobson & Welch, 1992*; *Muschick, Indermaur & Salzburger, 2012*). In particular, nitrogen and carbon stable isotopes have been found to reflect trophic position: both $\delta^{15}N$ and $\delta^{13}C$ tend to increase in a predictable manner with each successive trophic level (*Post, 2002*, but see *De Vries et al., 2015*). Stable isotopes have also been used to track nutrient flow, climatic shifts, and migration patterns in a variety of ecological systems (*Best & Schell, 1996*; *Chamberlain et al., 1997*; *Iacumin, Davanzo & Nikolaev, 2005*; *McMahon et al., 2016*). The current study uses stable isotopes of N and C to assess the extent to which entire food webs are influenced by the chemistry of their habitats. We chose to focus on spiders because they are mobile generalist predators and encompass multiple trophic levels, from feeding on herbivorous insects to specializing on the highest predator levels among Hawaiian arthropods (other spiders). This substantial variation in trophic ecology allows us to test how functional and trophic differences are reflected in isotopic signatures, and the extent to which the biogeochemistry of a food web is determined by the chemistry of the immediate substrate. The use of N and C stable isotopes is especially well suited to this study because of the predictable manner in which both elements can reflect trophic position, and because of the importance of nitrogen in ecosystem development (*Boring et al., 1988*).

We analyzed Hawaiian spiders belonging to two lineages within the adaptive radiation of long-jawed orb-weavers (*Tetragnatha*, Tetragnathidae) and one lineage within the stick spiders (*Ariamnes*, Theridiidae). The *Tetragnatha* radiation includes *ca.* 60 species, which display a spectacular array of colors, shapes, sizes, behaviors, and ecological affinities not observed elsewhere in the range of this genus (*Blackledge & Gillespie, 2004*; *Gillespie, 2004*; *Gillespie, 2015*). The radiation consists of two major clades: one that spins webs for prey capture ("web builders"), and another that has lost the web-spinning behavior and instead hunts actively (the "Spiny Leg" clade; *Gillespie, 1991*; *Gillespie, 2002*). Observational data indicate that both web-building and Spiny Leg *Tetragnatha* feed on a mixture of insect herbivores and predators (*Binford, 2001*), although the exact composition of these spiders' diets has not yet been fully characterized. The Hawaiian *Ariamnes*, currently represented by 11 known species across the Hawaiian Islands (*Gillespie & Rivera, 2007*), are also ecologically diverse and largely araneophagic (i.e., preying on other spiders) (*Gillespie et al., 2018*). Like *Tetragnatha*, these spiders are exclusively nocturnal, and like the Spiny Leg *Tetragnatha*, they hunt without the use of a web.

Given the contrasting hunting strategies (web-building versus active hunting) and trophic positions (generalist versus araneophagic) across these spider lineages, the three groups vary predictably in their position in the food web, from largely feeding on primary consumers (i.e., insect herbivores) to exclusively feeding on secondary and higher consumers (i.e., spiders). Within this system, we tested the hypothesis that isotopic signatures of spiders should reflect the biogeochemistry of their respective habitats, from young to older in the Hawaiian chronosequence. Thus, not only should the different spider

lineages illustrate now-standard expectations for isotope signatures associated with rising trophic levels, but the trophic ecology of the entire food web should reflect changes in soil chemistry across the chronosequence. In particular, given that $\delta^{15}$N in the soil increases during the building phase of the Hawaiian ecosystems (*Vitousek et al., 1997*), we expect $\delta^{15}$N to be lowest in the spiders at the youngest (200–750 y) site and highest in the spiders at the oldest (20,000 y) site.

## METHODS

### Study sites

Hawaii's sequential age structure has made it an ideal system for previous studies on soil evolution and nutrient cycling, wherein the Long Substrate Age Gradient (LSAG) was established (*Crews et al., 1995*; *Vitousek, 2004*). This study focuses on Hawai'i Island, the youngest in the archipelago, because the largest possible range of N availabilities is expected to be found there: soil nitrogen is lowest in the youngest substrates and peaks in the older substrates on Hawai'i Island before declining on the older islands (*Vitousek, Turner & Kitayama, 1995*).

Specimens were collected on the windward side of Hawai'i Island under permits from the State of Hawaii Department of Land and Natural Resources (endorsement # FHM14-349) and the National Park Service (study # HAVO-00425). Three sites of different substrate age, chosen for their comparable elevations and climates as well as the overlap of two of the sites ('Ola'a and Laupāhoehoe) with those characterized based on soils (*Vitousek et al., 1997*), were sampled (see Fig. 1). Substrate ages were determined based on data from the United States Geological Survey (*Sherrod et al., 2007*) and from the LSAG (*Crews et al., 1995*; *Vitousek, 2004*). All three sites are wet/mesic forest dominated by *Metrosideros polymorpha* and *Acacia koa* in the canopy, with *Cibotium* spp. dominating the understory. The sites range from 1,180 to 1,390 m in elevation, with mean annual temperatures of 13.9 to 15.4 degrees Celsius and mean annual rainfall of 3,035 to 3,090 mm (*Giambelluca et al., 2014*).

Upper Waiakea is a very young site on a 200- to 750-year-old lava flow in a stratified matrix of differently-aged substrates within the Upper Waiakea Forest Reserve, off of Stainback Highway on Mauna Loa. 'Ola'a Forest is on an older lava flow on Kilauea, situated within Hawai'i Volcanoes National Park. The trees in 'Ola'a are rooted in a thick layer of tephra of approximately 2,100 years old (*Vitousek, 2004*), beneath which is an older flow of 5,000–11,000 y (see Fig. 1). Although the United States Geological Survey (USGS) classifies this substrate as 5,000–11,000 y, we consider its biota to be influenced by the chemical properties of the 2,100-year-old tephra in which the forest is rooted. 'Ola'a is therefore approximately one order of magnitude older than Upper Waiakea, yet is located just 11.5 km S of the younger site. This proximity makes the two sites especially useful for measuring effects of habitat age on a small geographical scale. The oldest site in this study is Laupāhoehoe, located in the Laupāhoehoe Experimental Forest Unit on Mauna Kea. While USGS data (*Sherrod et al., 2007*) estimate the lava flow age at 5,000–11,000 y, the sampling locality overlaps with an LSAG site which has been studied in great detail and whose forest is rooted in a layer of soil believed to be approximately 20,000 y old (*Vitousek,*

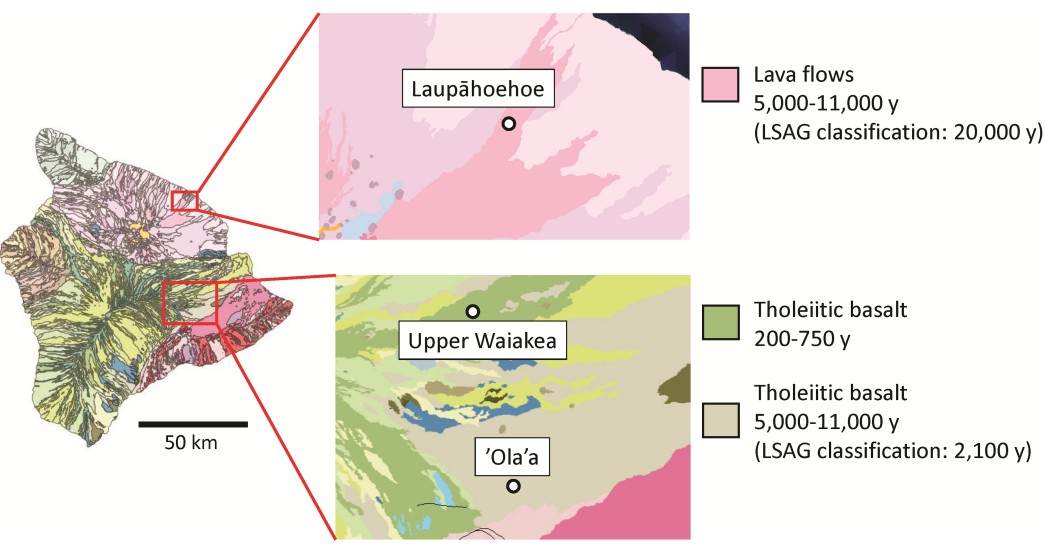

**Figure 1** **Map showing field sites where samples were collected.** Colors represent geology and lava flow age, determined by United States Geological Survey (*Sherrod et al., 2007*). Where applicable, substrate age classifications determined by the Long Substrate Age Gradient (LSAG, *Crews et al., 1995*) are included.

*2004*; *Vitousek et al., 2009*). We therefore follow the LSAG classification of 20,000 y for Laupāhoehoe.

Collections at each site were centered at the following coordinates, with searches extending up to 100 m in any direction:

- Upper Waiakea: 19.562°N, 155.272°W
- 'Ola'a: 19.462°N, 155.248°W
- Laupāhoehoe: 19.922°N, 155.301°W

## Specimen collection

Six species of *Tetragnatha* spiders (*Tetragnatha anuenue*, *T. brevignatha*, *T. hawaiensis*, *T. perkinsi*, *T. quasimodo* and the undescribed species *T.* "golden dome") and two species of *Ariamnes* spiders (*Ariamnes hiwa* and *A. waikula*) were collected in the field from 11 March to 18 April 2015, and from 6 to 15 February 2016. All study species are morphologically distinct and can be easily identified in the field. Plant samples were leaves of dominant or common forest vegetation (*Metrosideros polymorpha*, *Cibotium* spp. and *Cheirodendron* spp.), all of which are also easily identified on sight. After a preliminary analysis, it was determined that leaf litter should be added to the study in order to help explain differences in carbon isotope values. Unfortunately, permitting and time constraints only allowed for leaf litter to be collected from the youngest site (Upper Waiakea) on 15 October 2016.

Spiders were individually hand-captured into clean plastic snap-cap vials which were labeled on the outside with unique identifiers, while plant leaves were clipped off with scissors and stored in labeled paper envelopes. In Upper Waiakea, leaf litter was also collected from the bases of *M. polymorpha* trees and placed in paper envelopes. Leaves

were air-dried in their envelopes in a sealed container of silica for three weeks prior to processing. Leaf litter was dried in a 60 °C oven overnight.

Spiders were photographed up close using a Nikon D5200 with an AF-S DX Micro-NIKKOR 40 mm f/2.8 g lens and a Speedlight SB-400 flash from both dorsal and lateral angles. This created a photographic voucher and allowed for visual identification of species, sex, and maturity. Spiders were killed overnight in a freezer and air-dried in separate snap cap vials, each with one clean bead of silica gel, before being transferred to individual 2-mL centrifuge tubes.

## Stable isotope analysis

Individual dried spiders were weighed into 9 × 5 mm pressed tin capsules for isotopic analysis. To optimize N content for isotopic analysis, a 1.5-mg mass was recommended for each sample (S Mambelli, UC Berkeley Center for Stable Isotope Biogeochemistry, pers. comm., 2014). Due to the spiders' small body size (ranging from ca. 0.4 to 5 mg dry weight for the majority of specimens), it was not feasible to obtain sufficient material from individual body parts, although it has been found that different spider tissues can undergo different isotopic turnover rates (*Belivanov & Hambäck, 2015*). Therefore, in order to control for possible variation in isotopic turnover among tissue types, the spiders' entire bodies were analyzed. When spiders exceeded 2.5 mg dry weight, they were homogenized (powdered and mixed) with a mortar and pestle, and a 1.5-mg sample of the homogenized tissue was used. For smaller spiders (<2.5 mg), the whole intact body was packed into the tin capsule in order to avoid excessive loss of material. Plant leaves were individually homogenized in a Mini-Beadbeater (BioSpec Model 8) in 7-mL tubes with stainless steel ball bearings, then weighed into 9 × 5 mm tin capsules. Due to the relatively low N:C ratio of plants, 6 mg of material was weighed out for each leaf sample. Leaf litter was processed in the same manner as plant leaves.

Samples were analyzed for nitrogen and carbon content (% dry weight) and nitrogen and carbon stable isotope ratios via elemental analyzer/continuous flow isotope ratio mass spectrometry using a CHNOS Elemental Analyzer (model: Vario ISOTOPE cube; Elementar, Hanau, Germany) coupled with an IsoPrime 100 mass spectrometer (Isoprime Ltd, Cheadle, UK). The isotope ratio is expressed in "delta" notation (in parts per thousand, or ‰ units). The isotopic composition of a material relative to that of a standard on a per mill deviation basis is given by $\delta^{15}N$ (or $\delta^{13}C$) $= (R_{sample}/R_{standard} - 1) \times 1,000$, where $R$ is the molecular ratio of heavy to light isotopes. The standard for nitrogen is air. The standard for carbon is V-PDB. The reference material NIST SMR 1547 (peach leaves) was used as calibration standard (long-term precision [since 2000] using this standard is ±0.07‰ for both N and C isotope analyses). All isotope analyses were conducted at the Center for Stable Isotope Biogeochemistry at the University of California, Berkeley. Long-term external precision based on reference material NIST SMR 1577b (bovine liver) is 0.15‰ and 0.10‰, respectively, for N and C isotope analyses.

## Data analysis

Results from the isotopic analysis were categorized into the following functional groups: "plant" (foliar samples of the genera *Metrosideros*, *Cibotium* and *Cheirodendron*); "Spiny

**Table 1  2-way ANOVA results.** Results of 2-way ANOVA testing for effects of site, functional group, and site × functional group interaction on stable isotopes of samples. Significant effects are indicated in bold.

| Isotope | Effect | F | df | p-value |
|---|---|---|---|---|
| | Site | 692.1 | 2 | **<0.001** |
| $\delta^{15}N$ | Functional group | 113.6 | 4 | **<0.001** |
| | Site:functional group | 8.615 | 6 | **<0.001** |
| | Site | 55.51 | 2 | **<0.001** |
| $\delta^{13}C$ | Functional group | 95.15 | 4 | **<0.001** |
| | Site:functional group | 1.841 | 6 | 0.092 |

Leg" (Spiny Leg *Tetragnatha* species: *T. anuenue*, *T. brevignatha* and *T. quasimodo*); "web" (*Tetragnatha hawaiensis*, *T. perkinsi*, and the undescribed species nicknamed *T.* "golden dome"); and "*Ariamnes*" (*Ariamnes hiwa* and *A. waikula*, the two species found on Hawai'i Island (*Gillespie & Rivera, 2007*)). Although all spider species were initially analyzed separately, the results showed that the data grouped taxa together, largely in accordance with functional group. Because members of one group (e.g., web-builders) were closer to one another than to other groups (e.g., Spiny Leg), we chose to focus on these broader ecological categories—functional groups—rather than species.

Effects of site and functional group on $\delta^{15}N$ and $\delta^{13}C$ were tested using a 2-way Anova allowing for interaction (with site and functional group as factors) on R statistical software (version 3.2.2, 64-bit). Main effects were then analyzed separately: we tested (1) effect of site within each functional group and (2) effect of functional group within each site. When significant differences were found, pairwise comparisons were made using Tukey's honest significant difference test (*Tukey, 1949*).

# RESULTS

We found a significant interaction between site and functional group for $\delta^{15}N$ ($F = 8.615$, $p < 0.001$), but not for $\delta^{13}C$ ($F = 1.841$, $p = 0.092$; Table 1). Significant main effects were found for all variables tested: site for $\delta^{15}N$ ($F = 692.1$, $p < 0.001$), functional group for $\delta^{15}N$ ($F = 113.6$, $p < 0.001$), site for $\delta^{13}C$ ($F = 55.51$, $p < 0.001$), and functional group for $\delta^{13}C$ ($F = 95.15$, $p < 0.001$).

## Main effects: site

For $\delta^{15}N$, a significant site effect was found within all functional groups (plants: $F = 78.74$, $p < 0.001$; Spiny Leg: $F = 446.9$, $p < 0.001$; web-builders: $F = 216.6$, $p < 0.001$; *Ariamnes*: $F = 80.87$, $p < 0.001$; Table 2 and Fig. 2). We performed a Tukey's HSD test to reveal pairwise differences among sites. $\delta^{15}N$ showed a clear pattern of stepwise increase with substrate age (lowest in Upper Waiakea (200–750 y), intermediate in 'Ola'a (2,100 y), and highest in Laupāhoehoe (20,000 y)). This pattern held true for every functional group; the only comparison not found to be significant was 'Ola'a vs. Laupāhoehoe within *Ariamnes* (Tukey's adjusted $p = 0.203$).

For $\delta^{13}C$, a significant site effect was found within every functional group except for plants (plants: $F = 0.7997$, $p = 0.482$; Spiny Leg: $F = 5.681$, $p = 0.005$; web-builders:

**Table 2 Main effects of site within functional group.** Main effects of site (substrate ages: Upper Waiakea, 200–750 y; 'Ola'a, 2,100; Laupāhoehoe, 20,000 y) within each functional group of spiders and plants. Site was found to have a significant effect on both C and N isotope ratios of every functional group, with the exception of $\delta^{13}$C in plants.

| Isotope | Comparison | F | df | p-value |
|---|---|---|---|---|
| $\delta^{15}$N | Plants | 78.74 | 2 | **<0.001** |
| | Spiny Leg | 446.9 | 2 | **<0.001** |
| | Web-builders | 216.6 | 2 | **<0.001** |
| | *Ariamnes* | 80.87 | 2 | **<0.001** |
| $\delta^{13}$C | Plants | 0.7997 | 2 | 0.482 |
| | Spiny Leg | 5.681 | 2 | **0.005** |
| | Web-builders | 31.91 | 2 | **<0.001** |
| | *Ariamnes* | 36.62 | 2 | **<0.001** |

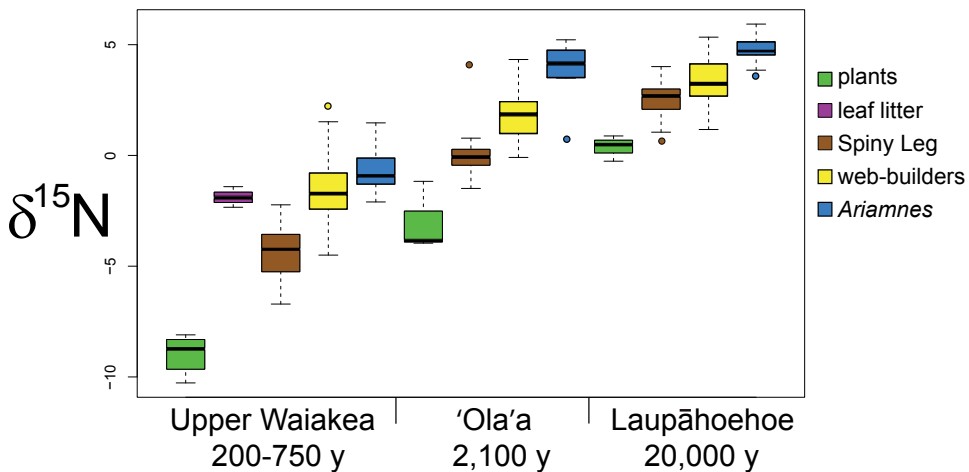

**Figure 2 Boxplots showing nitrogen isotope ratios of functional groups across sites.** Nitrogen isotope ratio ($\delta^{15}$N in ‰ units) of plant leaves (green), leaf litter (purple), Spiny Leg (brown), web-building (yellow) and *Ariamnes* (blue) spiders across sites of different ages.

$F = 31.91$, $p < 0.001$; *Ariamnes*: $F = 36.62$, $p < 0.001$; Table 2 and Fig. 3). Among the three groups of spiders, $\delta^{13}$C was significantly lower in Laupāhoehoe (20,000 y) than in 'Ola'a (2,100 y).

## Main effects: functional group

A significant functional group effect was found in all sites for $\delta^{15}$N (Upper Waiakea: $F = 68.38$, $p < 0.001$; 'Ola'a: $F = 34.23$, $p < 0.001$; Laupāhoehoe: $F = 28.90$, $p < 0.001$; Table 3 and Fig. 2). A Tukey's HSD test found significant differences among every pair of groups except for the following pairs in Upper Waiakea: web-builders vs. *Ariamnes* (Tukey's adjusted p-value = 0.071), web-builders vs. leaf litter (Tukey's adjusted p-value = 0.998), and *Ariamnes* vs. leaf litter (Tukey's adjusted p-value = 0.512).

For $\delta^{13}$C, a significant functional group effect was found at all three sites (Upper Waiakea: $F = 36.42$, $p < 0.001$; 'Ola'a: $F = 41.29$, $p < 0.001$; Laupāhoehoe: $F = 41.48$,

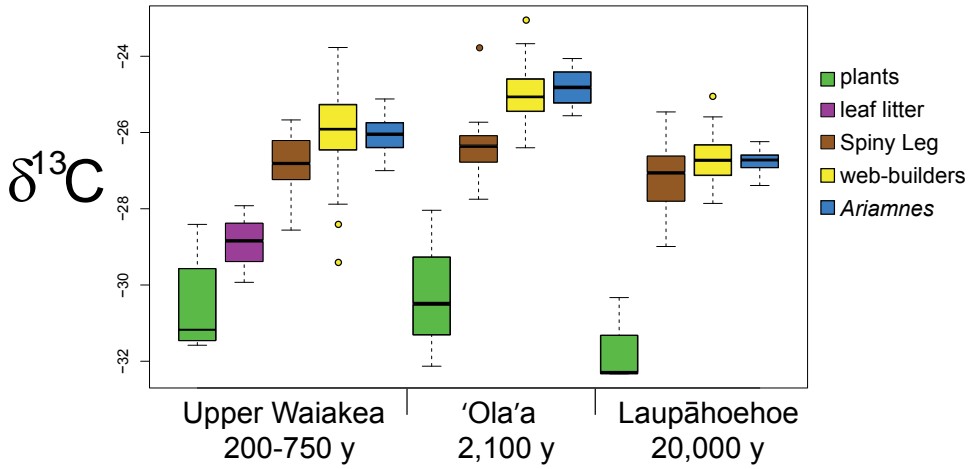

**Figure 3** **Boxplots showing carbon isotope ratios of functional groups across sites.** Carbon isotope ratio ($\delta^{13}C$ in ‰ units) of plant leaves (green), leaf litter (purple), Spiny Leg (brown), web-building (yellow) and *Ariamnes* (blue) spiders across sites of different ages.

**Table 3** **Main effects of functional group within site.** Functional groups were found to differ significantly from one another in their isotope ratios of both C and N at all three sites.

| Isotope | Site | F | df | p-value |
|---|---|---|---|---|
| $\delta^{15}N$ | Upper Waiakea (200–750 y) | 68.38 | 4 | **<0.001** |
| | 'Ola'a (2,100 y) | 34.23 | 3 | **<0.001** |
| | Laupāhoehoe (20,000 y) | 28.90 | 3 | **<0.001** |
| $\delta^{13}C$ | Upper Waiakea (200–750 y) | 36.42 | 4 | **<0.001** |
| | 'Ola'a (2,100 y) | 41.29 | 3 | **<0.001** |
| | Laupāhoehoe (20,000 y) | 41.48 | 3 | **<0.001** |

$p < 0.001$; Table 3 and Fig. 3). At all sites, plants were significantly lower in $\delta^{13}C$ than the next-lowest trophic level, Spiny Leg spiders (consumers). In 'Ola'a (2,100 y), Spiny Leg spiders had significantly lower $\delta^{13}C$ than either web-builders or *Ariamnes* (Tukey's adjusted $p$-value $< 0.001$). In Upper Waiakea (200–750 y), although there was no significant difference between *Ariamnes* and either of the *Tetragnatha* functional groups, the Spiny Leg *Tetragnatha* were significantly lower in $\delta^{13}C$ than web-builders (Tukey's adjusted $p$-value $< 0.001$). In Laupāhoehoe (20,000 y), no significant difference was found between any pair of spider groups.

# DISCUSSION

The results from this study provide a novel perspective on how changes in the substrate chemistry of the terrestrial land surfaces across a chronosequence of developing ecosystems are propagated up through an entire food web. To our knowledge, this is also the first study to characterize the isotopic signatures of different ecological groups represented by exemplary adaptive radiations of spiders in Hawaii.

## Parallel shifts in isotopic signature across the chronosequence of ecosystem development

Previous work on the ecological characteristics of forests across the chronosequence has detailed the evolution of Hawaiian ecosystems in the context of soil properties and vegetation (*Vitousek, Shearer & Kohl, 1989*; *Vitousek, Turner & Kitayama, 1995*; *Vitousek et al., 1997*). Our results show that the chemical signatures of nutrient availability that characterize a given site are borne all the way up to the highest trophic levels—top predators—on a Hawaiian chronosequence. Indeed, the $\delta^{15}$N values of spiders perfectly match expectations for their respective habitats. Where nitrogen is most limited—at the youngest site (Upper Waiakea)—spiders have the lowest values of $\delta^{15}$N; as substrate age and nitrogen availability increase, so too does the $\delta^{15}$N of spiders. This makes sense physiologically because when nitrogen is very limited, the lighter isotope ($^{14}$N) is not as easily lost in reactions, and is instead retained at a greater rate in an organism's tissues (*Austin & Vitousek, 1998*). Conversely, when biologically available nitrogen is very abundant, $^{14}$N is readily lost (e.g., in excretion), leaving behind a greater proportion of the heavier $^{15}$N in the organism's tissues. Thus, our results support the hypothesis that the isotopic signatures of the spiders—as well as the plants—track the changes in the geological age of the islands (*Sherrod et al., 2007*) and the associated changes in nitrogen in soils across the Hawaiian chronosequence measured by *Vitousek et al. (1997)*.

While it might not be surprising that the increase in $\delta^{15}$N in the spiders tracks the increases in plants across the geological gradient, the fact that the relationship is so tight is remarkable, as it suggests that even higher-level consumers (predators) reflect the $\delta^{15}$N of the immediate site. This result is especially notable because it was found in cursorial animals, which, because of their mobility, might be expected to show only a weak association with the substrates on which they were collected. Instead, the spiders carry clear signatures of their immediate ecosystem. Given that the sites that were sampled are in very close proximity (11.5 km between 'Ola'a and Upper Waiakea), and not separated by any significant physical barrier, the results suggest an extraordinary level of isolation among spider populations. This has implications for the mechanisms by which the *Tetragnatha* adaptive radiation may have arisen: Isolation between populations separated by short distances can serve as a crucible for evolution (*Carson, Lockwood & Craddock, 1990*). Perhaps the same mechanisms that are currently at work on Hawaii's youngest island also led to the rise of the approximately 60 endemic *Tetragnatha* species found across the archipelago today. Results for *Ariamnes* were similar, all higher than the other spider lineages, and values increasing with substrate age (though the increase from 'Ola'a to Laupāhoehoe was not significant, presumably due to a relatively small sample size ($n = 9$) at these sites). *Ariamnes,* like the *Tetragnatha*, has undergone a substantial, though smaller at 11–16 species, adaptive radiation across the islands (*Gillespie et al., 2018*).

The carbon isotope data show a less clear pattern than nitrogen, but nevertheless indicate site-specific differences among the spiders. Notably, the plant samples did not differ significantly in $\delta^{13}$C among the three sites, suggesting that perhaps $\delta^{13}$C does not accurately reflect nutrient differences among the substrates. By contrast, $\delta^{15}$N appears to strongly reflect nutrient availability at the different sites. However, spiders did consistently
show higher $\delta^{13}$C in 'Ola'a (2,100 y) than in Laupāhoehoe (20,000 y). Thus, the relationship between baseline (plant) signatures and higher predator (spider) signatures is weaker in carbon than in nitrogen. This suggests that spiders at the three sites may not be consuming exactly the same assemblages of prey, perhaps due to variations in the availability of different insect (potential prey) species at different sites. A detailed study of the precise compositions of these spiders' diets, using either molecular gut content analysis (e.g., *Krehenwinkel et al., 2017*) or an isotopic mixing model with robust sampling of the entire arthropod community, could greatly enhance our understanding of the processes that account for the differences in $\delta^{13}$C among spider populations.

## Trophic positions

Our stable nitrogen isotope data reflect the different functional roles and trophic positions of the Hawaiian spiders. Our results are consistent with the enrichment of the heavier isotope, $^{15}$N, at higher trophic levels, with plants having the lowest values of $\delta^{15}$N, *Tetragnatha* having intermediate values, and the spider-eating *Ariamnes* having the highest. Additionally, we found that the $\delta^{15}$N of the Spiny Leg (cursorial) spiders was consistently lower than that of web-building *Tetragnatha*. A simple explanation for the difference in $\delta^{15}$N in cursorial vs. web-building spiders may be that the different functional groups consume different prey. Cursorial spiders are likely to interact with abundant insect herbivores, while web-builders may trap a larger proportion of flying insects at higher trophic levels, such as hymenopteran or dipteran parasitoids, decomposers, or predators. This dietary difference has been used to explain the phenomenon of a higher $\delta^{15}$N in web-builders compared to cursorial spiders in a forest hedge community (*Sanders, Vogel & Knop, 2015*). Another possible explanation is that the difference is due to the manufacturing of the orb web itself. Given that webs are an excretory product, and that excretion tends to favor the lighter $^{14}$N, it may be that the higher levels of "excretion" lead to an enrichment in $^{15}$N in the bodies of web-builders compared with cursorial spiders. A number of previous studies suggest such an effect. For example, across a community of web-building riparian spiders, lower $\delta^{15}$N was found in *Miagrammopes* (Uloboridae; *Kelly, Cuevas & Ramírez, 2015*), a genus characterized by a reduced capture web (often just a single line; *Lubin, Eberhard & Montgomery, 1978*), than in other spiders. Likewise, a study of niche width across a guild of spiders showed that cursorial spiders consistently had the lowest $\delta^{15}$N, while orb web spiders had the highest (*Sanders, Vogel & Knop, 2015*). In each of these systems, the cursorial spiders showed the highest levels of intraguild predation (i.e., feeding on other spiders), indicating that trophic position itself is insufficient to explain the lower $\delta^{15}$N of the cursorial spiders relative to the web spinners. This observation raises the possibility again that the web spinning process itself leads to $\delta^{15}$N enrichment. However, further data are clearly needed to determine which of these explanations best accounts for the now recurring pattern of higher $\delta^{15}$N in web-builders compared with cursorial taxa.

The patterns we found in $\delta^{13}$C were less dramatic than in $\delta^{15}$N, but matched expectations. Foliar samples consistently had the lowest $\delta^{13}$C of all functional groups. Spider values were substantially offset from leaf values—approximately 4–5 per mill higher at all sites—which suggests a complex food chain consisting of many trophic levels below the spiders. This is

plausible given that spiders are obligate predators (secondary consumers), and therefore must be trophically removed from plants by at least two levels. The conventional wisdom with $\delta^{13}C$ is "you are what you eat" (*Hobson, Barnett-Johnson & Cerling, 2010*), meaning that most organisms are only slightly enriched in $^{13}C$ relative to their diets, with standard published offsets of less than 1 per mill for each successive trophic level (*Post, 2002*). Indeed, meta-analysis of isotopic studies has found an average discrimination factor of ~0.3 per mill from one trophic level to the next within invertebrates (*Caut, Angulo & Courchamp, 2009*), although it should be noted that many of the invertebrates included in that meta-analysis are aquatic, and no "standard" $\delta^{13}C$ offset for spiders or other terrestrial arthropods has yet been established. In an effort to fill the large gap between plants and spiders, we added samples of leaf litter from Upper Waiakea. The $\delta^{13}C$ of leaf litter fit neatly between leaves and spiders. This is to be expected given that the lighter $^{12}C$ is lost as respired $^{12}CO_2$ that is produced at a greater rate during decomposition, leaving the remaining litter relatively $^{13}C$ enriched (*Dawson et al., 2002*). Furthermore, this finding fits with previous studies of *Tetragnatha* trophic ecology, wherein it was observed that tipulid flies comprise a large proportion of the diet of *Tetragnatha* on Maui (*Binford, 2001*; *Blackledge, Binford & Gillespie, 2003*). Because tipulid larvae often feed on decomposing leaves (*Williams, 1942*), it is reasonable to surmise that tipulids' own $\delta^{13}C$ values fall close to those of the leaf litter, and that spiders on Hawai'i Island become relatively enriched in their $\delta^{13}C$ composition when feeding on these insects.

## Link between ecosystem properties and evolutionary processes

This study demonstrates that organisms at multiple tropic levels reflect the stoichiometric changes in soil across the geological chronosequence of the island, from very young (200–750 y) to older (20,000 y). The importance of this result is that it shows that the evolutionary processes associated with diversification are intimately linked to a landscape that, itself, changes through time. The detailed work of *Vitousek, Shearer & Kohl (1989)*, *Vitousek, Turner & Kitayama (1995)*, *Vitousek et al. (1997)* and *Vitousek et al. (2009)* documents the pattern of change in soil chemistry over extended time periods: Nitrogen and phosphorus increase almost linearly with time in the early stages of substrate development (up to 20,000 y for nitrogen and 150,000 y for phosphorus); this increase then levels off and declines on the oldest islands (4 my).

At the same time, it is now well established that organismal diversity increases over time during the early stages of formation of an island archipelago (*Whittaker, Triantis & Ladle, 2008*; *Lim & Marshall, 2017*), and that higher trophic levels depend on lower levels in island community assembly (*Simberloff & Wilson, 1970*), yet explanations for such patterns have, as yet, considered only area and age of the landscape. The documentation of peaks of diversity on middle aged islands of the Hawaiian Archipelago has been explained variously based on the interaction between age and area (*Gillespie & Baldwin, 2010*; *Lim & Marshall, 2017*). Notably missing from these studies is a link between evolutionary processes of diversification and shifts in nutrient availability associated with ecosystem succession. That organisms in Hawaii are intimately reflective of the ecosystem properties of their immediate habitat demonstrates that changes in nutrients associated with the island chronosequence

are propagated through trophic and functional groups of entire biological communities. While initial work has begun to address the ecosystem consequences of evolutionary change (*Elser et al., 2003*; *Laiolo et al., 2015*), this study provides preliminary insights into how ecosystem change may affect processes of evolution.

## CONCLUSIONS

Variation in $\delta^{15}$N data indicates that different spider lineages reflect their different functional roles and trophic positions in Hawaiian food webs, from those feeding largely on primary consumers to those feeding exclusively on secondary and higher consumers. Importantly, the relationships between these groups, in terms of their $\delta^{15}$N, remain strong across the chronosequence. Not only do the spiders' relative values of $\delta^{15}$N show the same pattern at each site, but their isotopic signatures also reflect the availability of nitrogen at different sites from younger to older ecosystems. The tight relationship between N availability, plant isotopic values, and spider isotopic values strongly suggests that the spiders are dispersal-limited and their populations are isolated from one another, even across short distances. Such isolation may be an important mechanism of speciation within the *Tetragnatha* adaptive radiation. This study shows that these evolving lineages of spiders are intimately associated with the properties of their ecosystem, which is also changing. The tight connection between the organisms and the characteristics of their substrate highlights the importance of considering the role of soil properties, particularly chemistry, in addition to age and area, to understand how biodiversity accumulates over time.

## ACKNOWLEDGEMENTS

We thank Stefania Mambelli and Wenbo Yang for their help with the stable isotope analyses. We also thank our colleagues on the Dimensions team for invaluable assistance with logistics in the field; the staffs of the Hawaii National Park Service and Natural Area Reserve System for help with permits; Charles Griswold and Alexandra Rueda Esteban for assisting with sample collection; and Natalie Graham for collecting leaf litter. We are deeply grateful to the reviewers and editors who provided helpful feedback on an earlier submission of this paper.

### Funding

This material is based upon work supported by the National Science Foundation Graduate Research Fellowship under Grant No. DGE 1106400. This research was supported in part by the Margaret C. Walker Fund for teaching and research in systematic entomology. Additional funding was provided by the Robert L. Usinger Memorial Award and the Harvey I. Magy Memorial Scholarship Award. There was no additional external funding received for this study. The funders had no role in study design, data collection and analysis, decision to publish, or preparation of the manuscript.

## Grant Disclosures

The following grant information was disclosed by the authors:

National Science Foundation Graduate Research Fellowship: DGE 1106400.

Margaret C. Walker Fund.

Robert L. Usinger Memorial Award.

Harvey I. Magy Memorial Scholarship Award.

## Competing Interests

The authors declare there are no competing interests.

## Author Contributions

- Susan R. Kennedy conceived and designed the experiments, performed the experiments, analyzed the data, prepared figures and/or tables, authored or reviewed drafts of the paper, approved the final draft.
- Todd E. Dawson conceived and designed the experiments, contributed reagents/materials/analysis tools, authored or reviewed drafts of the paper, approved the final draft.
- Rosemary G. Gillespie conceived and designed the experiments, contributed reagents/materials/analysis tools, authored or reviewed drafts of the paper, approved the final draft.

## Field Study Permissions

The following information was supplied relating to field study approvals (i.e., approving body and any reference numbers):

Field work was approved by the State of Hawaii Department of Land and Natural Resources (endorsement number FHM14-349) and the National Park Service (study number HAVO-00425).

## Data Availability

Data will be available at UC Berkeley's Essig Musuem database (https://essigdb.berkeley.edu/) as well as at: Kennedy, Susan R, Dawson, Todd E, & Gillespie, Rosemary G. (2018). Raw data: Stable isotopes of Hawaiian spiders reflect substrate properties along a chronosequence [Data set]. PeerJ. Zenodo. http://doi.org/10.5281/zenodo.1194482.

## Supplemental Information

Supplemental information for this article can be found online at http://dx.doi.org/10.7717/peerj.4527#supplemental-information.

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
