# Peer review of "Stable isotopes of Hawaiian spiders reflect substrate properties along a chronosequence"

_PeerJ, doi:10.7717/peerj.4527_

## Round 0.1 · original submission · Minor Revisions

Dear Susan,

Your paper have been revised by two reviewers that were in favor of accepting your manuscript after minor revisions.

It is important to mention that one of the reviewers already reviewed a previous version of your manuscript, and it is the opinion of this reviewer that you have addressed well all his/her main concerns raised before.

Based on that, and after reading your manuscript I am confident that after taking care of minor revisions your manuscript will be accepted by PeerJ.

Best regards,

Luiz Martinelli

Reviewer 1 ·

Basic reporting

The manuscript is written in English using clear text and and appropriate structure.

Experimental design

The manuscript meets the aims and scope of the journal and the hypotheses are clearly stated. My largest concern about the paper is about the sampling design. There are no replicas of each chronosite (Upper Waiakea, ‘Ola’a and Laupāhoehoe). In addition:

Lines 215-216: I suggest to describe in more detail the laboratory procedures. Different parts of the spider's body may show variations in isotopic compositions. In addition, were gut contents removed before grinding spiders?

Lines 242-244: The criteria for grouping the isotopic samples are extremely important because they directly influence the results of the study. I suggest the inclusion of a scatter plot for each functional group at each collection site so that the variation of the data can be verified.

Validity of the findings

This manuscript demonstrates how the biogeochemistry of the food web is determined by ecosystem succession of the substrates on which the organisms have evolved. In this case, the data strongly show that the isotope values of the organisms reflect the chronosequence.

Additional comments

The manuscript entitled “Stable isotopes of Hawaiian spiders reflect substrate properties along a chronosequence” by Kennedy et al. is very interesting and presents important information concerning how different functional groups of spiders reflect the successional changes in the soil stoichiometry of the island chronosequence. This manuscript was previously submitted to Ecography and has been revised following the suggestions of the reviewers. The authors answered adequately the great majority of the questions asked by the reviewers. This fact made it possible to improve the manuscript.
In my opinion, the manuscript can be accept for publication since some essential minor revisions.

Reviewer 2 ·

Basic reporting

no comment

Experimental design

no comment

Validity of the findings

The authors could describe their one negative result in more detail in the Discussion. See below:

Line 277-282: This is true except for Ola’a vs. Laupāhoehoe within Ariamnes, as was mentioned in the Results (Line 244), correct? This neutral result is interesting. Can the authors elaborate on it more in the Discussion? I think there is a great opportunity to contrast Arimnes with Tetragnatha at the end of Line 301, as you have just described the implications for the results with respect to Tetragnatha. But what do the results mean for Ariamnes, given the ecology of Ariamnes?

Additional comments

The manuscript is much improved and clearer from the versions that I read from the previous submission to Ecography. The authors reframed the manuscript such that many of my initial concerns have been addressed. Below are a few more suggestions and questions about the study.

In the Discussion, it would be nice to emphasize that the results of this study are even more exciting because they were found in mobile predators. One could argue that the chronosequence would be more clear in more sedentary, herbivorous organisms and thus, seeing them in mobile predators makes this finding even more compelling. This was touched upon a bit in the Introduction, but it might be nice to emphasize in the Discussion as well, if space permits.

Line 69: Consider saying something along the lines of “disappearing” as opposed to “collapsing”. I found “collapsing” to be confusing.

Line 105: Change “one” to “within” so it reads “within stick spiders.”

Line 106-119: There is much more information about the Tetragnatha group compared to the Ariamnes group. Is all of the information about this group necessary? It seems like the relevant information is that some are web builders feeding on a mixture of insect herbivores and predators while others are cursorial predators.

Lines 302-312: I t might be worth explicitly stating that d15N seems to be reflecting nutrient differences while d13C in plants does not.

Lines 345-348: Are there published trophic discrimination factors for spiders? If so, it would be worth including them here. Post 2002 is also based on aquatic food webs. Citing a terrestrial study on invertebrates along with Post 2002 would also be useful.

External reviews were received for this submission. These reviews were used by the Editor when they made their decision, and can be downloaded below.

---

## Round 0.2 · accepted · Accept

I read the rebuttal letter and the revised manuscript. It is my opinion that the authors have addressed all the reviewers suggestions and this manuscript can be now accepted. Congratulations!

External reviews were received for this submission. These reviews were used by the Editor when they made their decision, and can be downloaded below.